# Conserving Ecosystem Diversity in the Tropical Andes

Patrick J. Comer [1,*], Jose Valdez [2,3], Henrique M. Pereira [2,3], Cristina Acosta-Muñoz [4], Felipe Campos [5], Francisco Javier Bonet García [4], Xavier Claros [6], Lucia Castro [7], Franciscio Dallmeier [8], Enrique Yure Domic Rivadeneira [9], Mike Gill [10], Carmen Josse [11], Indyra Lafuente Cartagena [6], Roberto Langstroth [12], Daniel Larrea–Alcázar [6,13], Annett Masur [2,3], Gustavo Morejon Jaramillo [11], Laetitia Navarro [2,14], Sidney Novoa [7], Francisco Prieto-Albuja [5], Gustavo Rey Ortíz [15], Marcos F. Teran [6], Carlos Zambrana-Torrelio [16] and Miguel Fernandez [2,3,16]

1   NatureServe, Boulder, CO 80305, USA
2   German Centre for Integrative Biodiversity Research (iDiv) Halle-Jena-Leipzig, Puschstrasse 4, 04103 Leipzig, Germany; jose.valdez@idiv.de (J.V.); hpereira@idiv.de (H.M.P.); annett.masur@idiv.de (A.M.); miguel.fernandez@idiv.de (M.F.); laetitia.navarro@ebd.csic.es (L.N.)
3   Institute of Biology, Martin Luther University Halle Wittenberg, Am Kirchtor 1, 06108 Halle (Saale), Germany
4   Department of Botany, Ecology and Plant Physiology, University of Cordoba. C.U. Rabanales, 14014 Cordoba, Spain; cristina.acosta@uco.es (C.A.-M.); fjbonet@uco.es (F.J.B.G.)
5   Instituto Nacional de Biodiversidad INABIO, Pasaje Rumipamba 341 y Av. de los Shyris, Quito 170505, Ecuador; felipe.campos@biodiversidad.gob.ec (F.C.); francisco.prieto@biodiversidad.gob.ec (F.P.-A.)
6   Asociación Boliviana para la Investigación y Conservación de Ecosistemas Andino-Amazónicos (ACEAA-Conservación Amazónica), Calacoto Calle 16 #8230, La Paz, Bolivia; xclaros@conservacionamazonica.org.bo (X.C.); ilafuente@conservacionamazonica.org.bo (I.L.C.); dlarrea@conservacionamazonica.org.bo (D.L.-A.); mteran@conservacionamazonica.org.bo (M.F.T.)
7   Asociación para la Conservación de la Cuenca Amazónica—ACCA, Calle Vargas Machuca 627, Miraflores, Lima 15047, Peru; lcastro@conservacionamazonica.org (L.C.); snovoa@conservacionamazonica.org (S.N.)
8   Center for Conservation and Sustainability, Smithsonian Conservation Biology Institute, 1500 Remount Road, Front Royal, VA 22630, USA; dallmeierf@si.edu
9   Independent Researcher, Calle Pedraza #344 Zona Irpavi, La Paz, Bolivia; eydomic@gmail.com
10  NatureServe, 2550 South Clark St. Suite 930, Arlington, VA 22202, USA; mike.gill@natureserve.org
11  Fundación EcoCiencia, San Ignacio E12-143 y Humboldt, Edificio Carmen Lucia Department 1 (Sector González Suárez—Norte de Quito), Quito 170517, Ecuador; carmenjosse@ecociencia.org (C.J.); gustavo@save.bio (G.M.J.)
12  Independent Researcher, 43611 Hetrick Ln, South Riding, VA 20152, USA; pampa_isla@yahoo.de
13  Herbario Nacional de Bolivia (LPB), Instituto de Ecologia (IE), Universidad Mayor de San Andres (UMSA), Casilla 10077, Correo Central, La Paz, Bolivia
14  Departamento de Biología de la Conservación, Estación Biológica de Doñana, Américo Vespucio n° 26, 41092 Sevilla, Spain
15  Museo Nacional de Historia Natural, La Paz, Bolivia; greyortiz@mnhn.gob.bo
16  Department of Environmental Science and Policy, College of Science, George Mason University, 4400 University Drive, Fairfax, VA 22030, USA; cmzambranat@gmail.com
*   Correspondence: pat_comer@natureserve.org

**Abstract:** Documenting temporal trends in the extent of ecosystems is essential to monitoring their status but combining this information with the degree of protection helps us assess the effectiveness of societal actions for conserving ecosystem diversity and related ecosystem services. We demonstrated indicators in the Tropical Andes using both potential (pre-industrial) and recent (~2010) distribution maps of terrestrial ecosystem types. We measured long-term ecosystem loss, representation of ecosystem types within the current protected areas, quantifying the additional representation offered by protecting Key Biodiversity Areas. Six (4.8%) ecosystem types (i.e., measured as 126 distinct vegetation macrogroups) have lost >50% in extent across four Andean countries since pre-industrial times. For ecosystem type representation within protected areas, regarding the pre-industrial extent of each type, a total of 32 types (25%) had higher representation (>30%) than the post-2020 Convention on Biological Diversity (CBD) draft target in existing protected areas. Just 5 of 95 types (5.2%) within the montane Tropical Andes hotspot are currently represented with >30% within the protected areas. Thirty-nine types (31%) within these countries could cross the 30% CBD 2030 target with the addition

of Key Biodiversity Areas. This indicator is based on the Essential Biodiversity Variables (EBV) and responds directly to the needs expressed by the users of these countries.

**Keywords:** essential biodiversity variables; ecosystem loss; EBV cube; ecosystem representation; trends in extent; protected areas; key biodiversity area; GEO BON

## 1. Introduction

The Tropical Andes is one of the most important biodiversity hotspots in the world. With less than 0.5% of the Earth's land surface, it hosts ten percent of all known species, as well as the most endemic plants and vertebrate species in the world [1]. This is due to its large spatiotemporal variability of environmental conditions which support over one hundred unique types of described terrestrial ecosystems [2,3]. Despite its global importance, the Tropical Andes is also one of the most severely threatened areas and, having lost at least one-quarter of its original extent to intensive land uses [1,3–9], is a key priority for conservation [1,4,10]. This region will not only continue to undergo severe stresses from human activities over the next century, but its diverse natural habitats are extremely vulnerable to climate change [2,11,12], with more than half of the species expected to undergo range reductions and 10% of the species becoming extinct by 2050 [13]. Protecting the Tropical Andes from anthropogenic threats and reducing its rate of habitat loss is an urgent priority for conservation and research efforts [14].

The single most important conservation strategy for the Tropical Andes has been the establishment of natural protected areas [15]. However, 72% of all species and 90% of threatened endemic species are insufficiently covered [16], with protected areas no more representative of biodiversity than nonprotected areas [17]. Additionally, 77% of the protected areas in the Tropical Andes are located in places that are less vulnerable to habitat change and exhibit low irreplaceability [16], which has resulted in conservation goals failing for more than half of all species in the region [18]. One solution is the inclusion of Key Biodiversity Areas (KBAs) that could be targeted to conserve globally significant—often most irreplaceable—components of biological diversity. Criteria for classifying KBAs include locations supporting threatened species and ecosystems, geographically restricted biodiversity, landscapes of high ecological integrity, critical biological processes, and/or areas identified as highly irreplaceable through quantitative analysis. KBAs can therefore play a role in representing ecosystem diversity within broader conservation strategies.

Protecting the full range of ecosystem diversity, including the most rare and vulnerable ecosystems, must be a priority when designating new sites for future protected areas. Because natural ecosystem patterns and processes influence the composition of communities in the short run and define selective pressures on organisms over evolutionary time frames, the loss of habitat extent would be expected to correlate with a decline in niche diversity, species diversity, and variability in key ecological processes [19,20]. Tracking progress on ecological representativeness of protected area networks remains an important indication of progress in biodiversity conservation.

Under the Convention on Biological Diversity (working draft) Strategic Plan Post-2020 Framework [21], a series of targets have been proposed to support conservation action and monitor progress. Among the Plan's action targets for 2030, Target 3 is to "Ensure that at least 30 percent global land area, are conserved through effectively and equitably managed, ecologically representative and well-connected systems of protected areas." The meaning of "ecologically representative" has been interpreted variously, with one common interpretation being the surface area of ecoregions [22].

In 2013, the Group on Earth Observations—Biodiversity Observation Network (GEO BON) proposed the concept of Essential Biodiversity Variables (EBVs), consisting of a data cube with three basic dimensions (taxonomy, time, and space). The initial list of EBVs was defined as the key measurements required to study, report, and manage the

multiple dimensions of biodiversity change [23]. These variables are intended to serve as a bridge between primary biological observations and summary indicators for use by policymakers and must fulfill criteria on scalability, temporal sensitivity, feasibility, and relevance. Maps of classified ecosystem types of varying levels of thematic detail should serve as the basis for the ecosystem extent EBVs which fall under the ecosystem structure of the EBV framework. However, beyond being able to measure trends in extent, which is already a useful element for decision-makers, these high-resolution ecosystem maps allow us to measure more subtle changes in the functional, structural, and compositional aspects of ecosystem dynamics by providing hierarchical spatial data.

To complement the development of EBVs, Navarro et al. [24] proposed a process to develop sustained, user-driven, locally operated, harmonized, and scalable Biodiversity Observation Networks (BONs), which was implemented for the establishment of the Tropical Andes Biodiversity Observation Network (TAO). The first step in the development of TAO was to assess user needs through a national and regional stakeholder consultation process. This consultation process was carried out with different communities of biodiversity information users in Ecuador, Peru, and Bolivia during 2019 and 2020, and identified the need for an EBV-based indicator that can help measure trends in the extent of ecosystems and their degree of conservation in the region.

However, this demand is not easy to meet because the terrestrial ecosystems of the Tropical Andes are extremely difficult to observe with traditional methods due to limited accessibility, complex topography, and constant cloud cover. Our objective is to use remote sensing data and analytical tools that facilitate the assessment of trends in ecosystem extent and the proportional area protected building on the results of recent mapping efforts with hierarchical vegetation classification [25,26]. This analysis should form a practical foundation for trend assessment of terrestrial ecosystems that can support countries to measure the effectiveness and impact of their national policies, as well as to report progress toward multilateral agreements and commitments (e.g., Ramsar, SDGs, and Post-2020 Global Biodiversity Framework). Furthermore, the nested organization of units in this ecosystem classification and the 270 m pixel resolution provide enough detail to be used as a reference for implementing conservation actions at the departmental and municipal levels and the flexibility necessary to be easily amenable to reporting trends at the national and continental scales.

## 2. Materials and Methods

### 2.1. Study Area

This analysis focused on the Tropical Andes region of Colombia, Peru, Ecuador, and Bolivia (Figure 1). This is a region with an extreme elevational range with vertical distances of more than 6500 m in less than 150 km of horizontal distances, representing an extremely diverse combination of ecotones and life zones that harbors unique combinations of species richness, endemism, and threat, which have all contributed to its recognition as a global biodiversity hotspot [1].

This area spans latitudes from 15°53′ N in Colombia to 22°54′ S in Bolivia and altitudes from sea level to 6768 m at Mount Huascarán in Peru. The project area includes approximately 3.8 million km$^2$. We report findings with all four countries combined, within individual countries, and within the Tropical Andes montane hotspot.

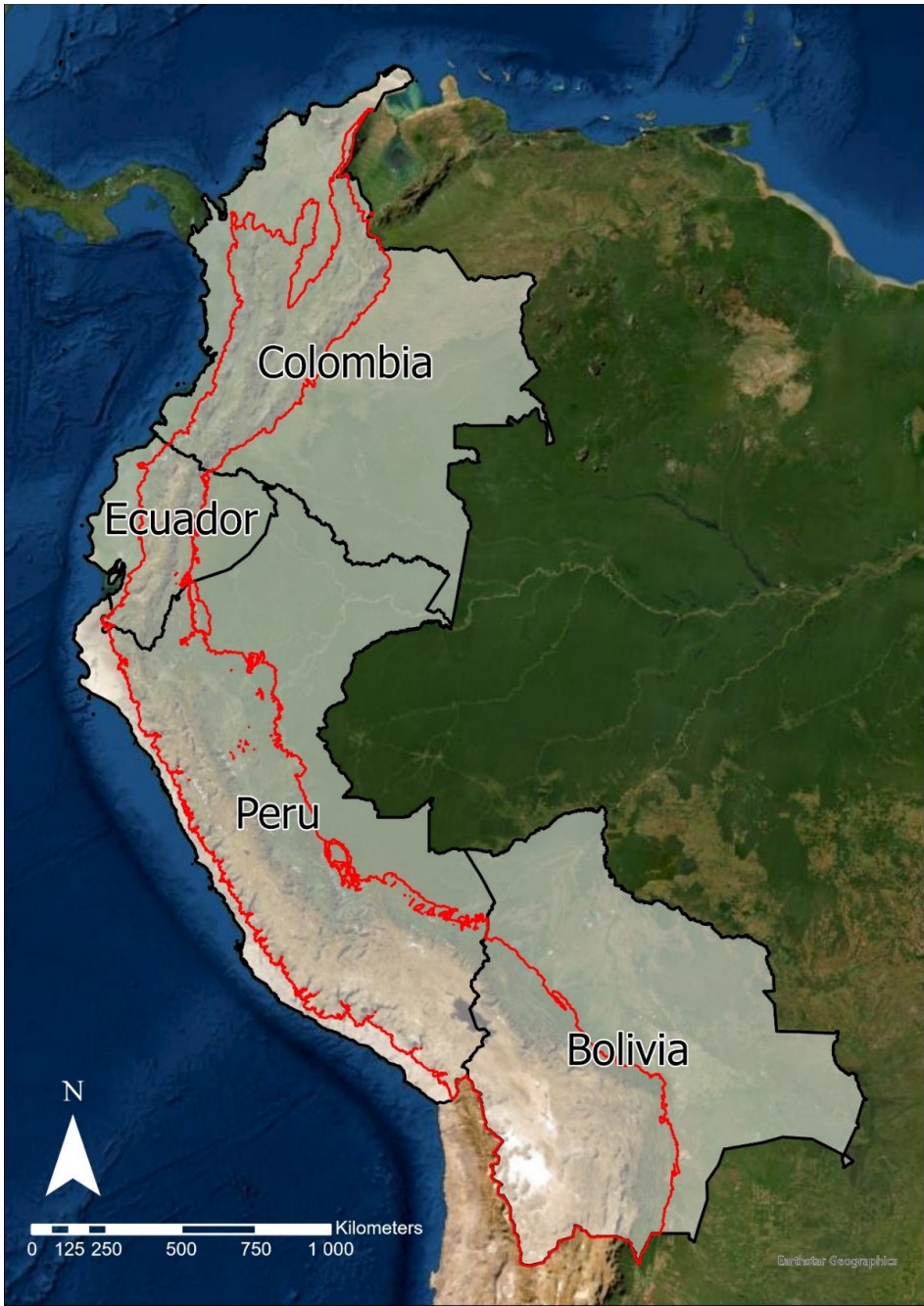

**Figure 1.** Study area encompassing Tropical Andes countries and the associated biodiversity hotspot (in red; Source: Critical Ecosystem Partnership Fund).

*2.2. GEOBON User Needs Assessment*

From 2019 through 2022, several local institutions in the Tropical Andes, including Conservacion Amazonica-Peru (ACCA), Asociacion Boliviana para la Investigacion de Ecosistemas Andino Amazonicos (ACEAA), Fundacion Ecociencia, Ecuador, Instituto Nacional de Biodiversidad de Ecuador (INABIO), and several international institutions including NatureServe, Universidad de Cordoba in Spain, the Global Biodiversity Information Facility (GBIF), the German Centre for Integrative Biodiversity Research (iDiv), and the Group on Earth Observations—Biodiversity Observation Network (GEO BON) worked together to understand the needs of biodiversity data users in the Tropical Andes. The process began with national inventories of indicators and assessment, and a stakeholder consultation that reached over 400 biodiversity information generators and over 300 biodiversity users from

Bolivia, Ecuador, and Peru. This process was followed in 2021 by a multiday workshop at the regional level in which these national needs were assessed in the regional context. In 2022, a BON design workshop organized by iDiv was held to synthesize the needs into four major areas that will benefit directly from EBV-based indicators: risk management, land-use planning, major development projects, and infrastructure and biodiversity use by local communities. This process led to the prioritization of several indicators from which changes in the extent of ecosystems, and their current level of protection, is a critical indicator to be able to respond directly to the needs at the level of the Tropical Andes hotspot.

### 2.3. Targeted Ecosystems

Natural terrestrial ecosystem types were defined at the macrogroup level of the International Vegetation Classification (IVC) [27]; which had been mapped continent-wide [25,26] (Table 1). The IVC can be used to define the target map legends at multiple levels of detail [26]. The eight-level hierarchical structure of this classification follows that established as a federal standard for vegetation description in the United States, with broad units at upper levels defined by vegetation physiognomy, followed by progressively narrower units at lower levels defined by vegetation floristic composition.

**Table 1.** International Vegetation Classification Hierarchy, including example names of classification units from the Tropical Andes. The number of natural types documented (as of 2022) within each hierarchical level from Latin America and the Caribbean (* *classification incompletely developed at lower levels*).

| Level No. | Level Name | Defining Characteristics | No. Types | Example (Name) |
|---|---|---|---|---|
| 1 | Class | Life Form Physiognomy | 6 | Forest and Woodland |
| 2 | Subclass | Global Physiognomy | 13 | Tropical Forest and Woodland |
| 3 | Formation | Global Physiognomy | 32 | Tropical Montane Humid Forest |
| 4 | Division | Continental Floristics | 98 | Tropical Andean Montane Humid Forest |
| 5 | Macrogroup | Subcontinental Floristics | 274 | Northern Andean Montane and Upper Montane Humid Forest |
| 6 | Group | Regional Floristics | 814 * | Bosques Altimontanos Norte-Andinos de *Polylepis* |
| 7 | Alliance | Local Physiognomy and Floristics | * | * |
| 8 | Association | Local Floristics | * | * |

Within the IVC hierarchy, the vegetation macrogroup is level 5 of the 8 levels and would be viewed as a middle level of classification in terms of thematic detail suitable for mapping at regional to continental scales [27]. While the IVC encompasses the full spectrum of "natural" to "cultural" vegetation, here we focused solely on types considered to be "natural".

Through spatial modeling, Comer et al. [25] mapped both "potential" (i.e., pre-industrial land use) and "recent" distribution for IVC macrogroups (Figure 2). The "potential distribution" includes biophysical conditions where each type might occur today had there not been any prior intensive human intervention (i.e., since circa 1500). "Recent" distribution then indicates areas of intensive intervention and conversion, as of approximately 2010. For that effort, the mapped pixel resolution was 270 m. Comer et al. [25] produced a composite map for current land use (circa. 2010) for all of South America by combining products GlobCover (270 m pixel resolution, ca. 2009), and GlobeLand30 (30 m pixel resolution circa 2000–2010) [28]. Mapping methods and validation statistics are detailed in [25,26]. Map data may be accessed through [25]. Briefly, spatial modeling used georeferenced samples that had been labeled to each type and used as predictors to map surfaces reflecting climate, landform, and soils to depict a "potential" distribution of types on the target legend.

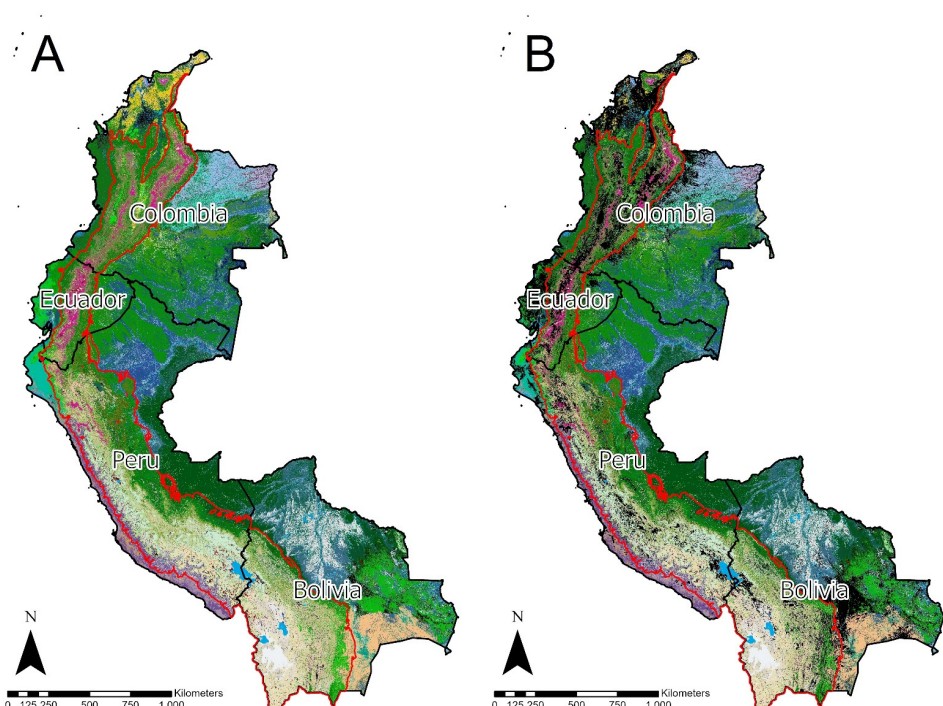

**Figure 2.** Pre-industrial (**A**) and recent (**B**) extent of 126 IVC macrogroups in the countries of the Tropical Andes. NOTE: black in (**B**) = human land use.

A total of 126 IVC macrogroups were mapped in the Tropical Andes countries, with 95 specifically within the Tropical Andes montane hotspot (Figure 2). The formations (IVC level 3) with the highest potential extent were forest to open woodlands, particularly tropical lowland humid forests, tropical flooded and swamp forests, tropical montane humid forests, and tropical high montane scrubs and grasslands. See links to type descriptions for IVC macrogroups included in Supplementary Materials S1.

### 2.4. Measuring Long-Term Type Loss from Land Conversion

We clipped the layer from Comer et al. [25] to our targeted countries' boundaries and then combined it with the potential distribution map of IVC macrogroups to indicate the current extent of macrogroups and land use classes ca. 2010 for our entire study area. Therefore, areas where current land use classes overlapped with natural ecosystem types from the potential distribution map were presumed to have been converted from natural ecosystem type to current land-use classes. This map combination resulted in an estimate of the extent of loss for each vegetation macrogroup from the pre-industrial, or "potential" extent, to the "recent" (circa. 2010) extent.

### 2.5. Ecosystem Type Representation in Protected Areas (IUCN I-VI)

To calculate/estimate the proportion of each macrogroup in existing protected areas, we assessed both pre-industrial and recent distributions occurring within natural protected areas. We accessed the World Database of Protected areas on 31 January 2022 for these data. The IUCN has established a globally applicable measure of conservation land status that includes 6 protected area categories [29] (Figure 3). These six categories range from Category I representing "Strict Nature Reserve" to Category VI representing "Protected area with sustainable use of natural resources". We did not compare this global data set to country-specific protected area data sets for this analysis.

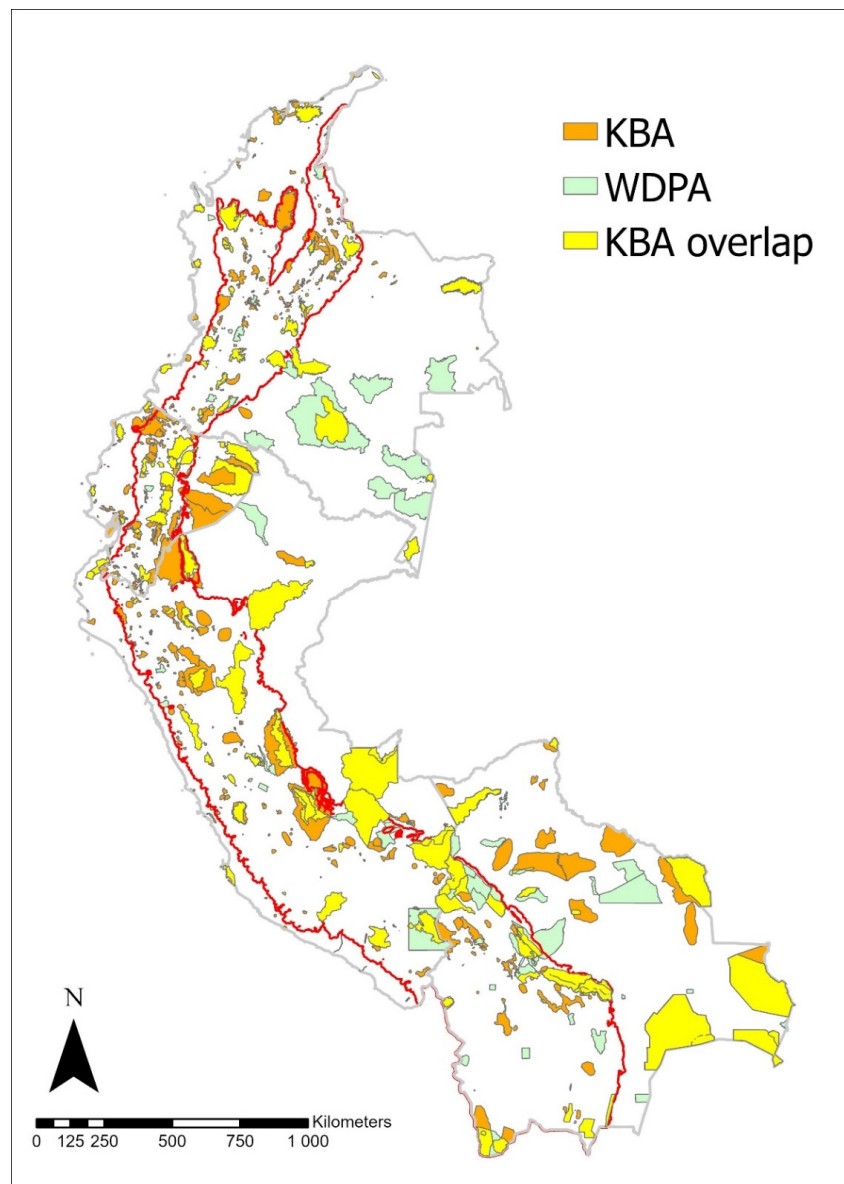

**Figure 3.** Key Biodiversity Areas (KBA), IUCN I-VI Protected areas from the World Database of Protected Areas (WDPA), and areas where KBAs overlap with existing Protected areas within countries encompassing the Tropical Andes hotspot (hotspot boundary in red).

We then visualized these per-type protection estimates by applying the calculated proportion of potential extent protected to the potential distribution maps to depict distributions in terms of relative protection classes (>50% protected down to <10% protected). Of course, we would not presume that representation within protected areas was truly "protected" in that actual management within and among protected areas determines much about the degree to which these ecosystems are being conserved.

### 2.6. Additional Ecosystem Representation in Key Biodiversity Areas

We then calculated the additional proportions of each IVC macrogroup (based on both pre-industrial and recent extents) that could be secured through identified Key Biodiversity Areas within the Tropical Andes countries (Figure 3). Here, we assessed proportions occurring in any identified KBAs to initially document ecosystem representation gained from these lands.

*2.7. Utilizing Multiple Levels of the IVC Classification Hierarchy*

We conducted a parallel spatial analysis a level up and level down from the macrogroup of the IVC hierarchy. For applications to continental or global analysis, we rolled up results to the IVC formation level. We also investigated ecosystem patterns below the IVC macrogroup level to discuss how macrogroup-level summaries might be stepped down to local conservation applications.

**3. Results**

*3.1. Long-Term Loss in Extent—Vegetation Macrogroups of the Tropical Andes*

Within the Tropical Andes countries, long-term loss in extent was 20.6% of all vegetation macrogroups combined in the four countries, with 43.9% loss within the Tropical Andes hotspot (Supplementary Materials S2a). Most of the loss of ecosystem extent has occurred across the Northern Andes in Colombia, Ecuador, and western Peru; with habitat loss in Eastern Bolivia as well (Figure 4). Six macrogroups had loss estimates >50% of their pre-industrial extent, including the Southern Andean Montane Salt Marsh (87.5%), the Guajiran Seasonal Dry Forest (60.6%), the Guajiran Xeromorphic Scrub and Woodland (59.6%), the Western Ecuadorian Humid Forest (57.3%), the Guajiran Flooded Forest (56.4%), and the Mesoamerican Floodplain Forest (52.7%) (see details in Supplementary Materials S2a).

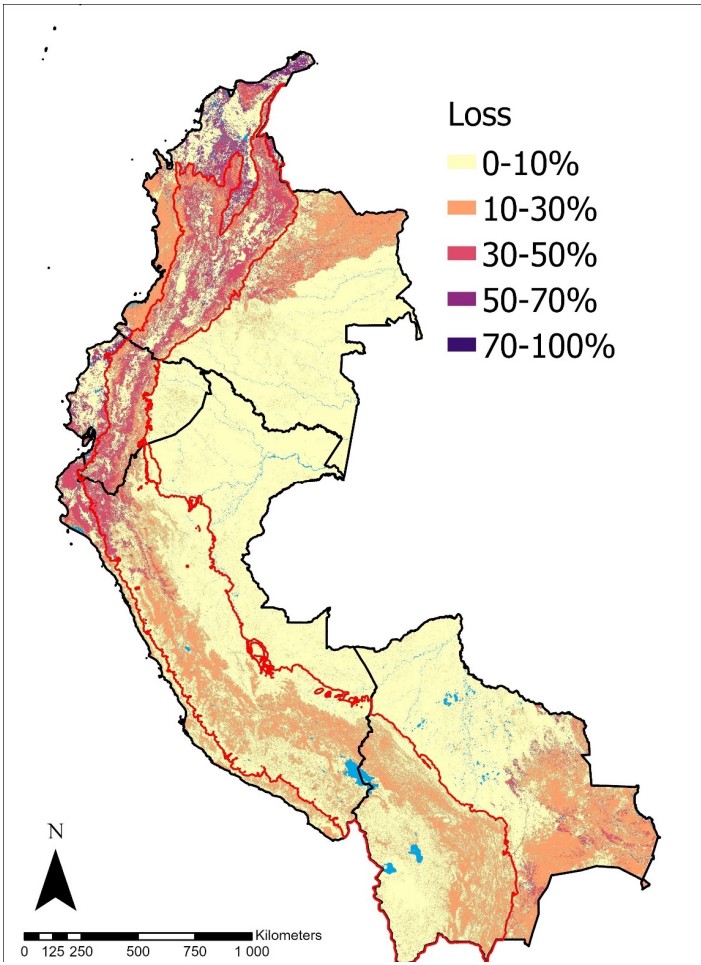

**Figure 4.** Long-term proportional loss in extent by IVC macrogroup (n = 126) (blue = water bodies).

Table 2 includes areal extent loss for 22 macrogroups summarized by Tropical Andes countries and the Tropical Andes hotspot within each country. All macrogroups listed in Table 2 have lost >50% of their historical extent in at least one of these geographic subset areas.

**Table 2.** Long-term loss in extent for IVC macrogroups summarized by country and within the Tropical Andes hotspot portion of each country. Those with >50%, >70%, and 100% loss are highlighted.

| IVC Macrogroup | Bolivia % Loss | Bolivia % hotspot Loss | Colombia % Loss | Colombia % Hotspot Loss | Ecuador % Loss | Ecuador % hotspot Loss | Peru % Loss | Peru % Hotspot Loss |
|---|---|---|---|---|---|---|---|---|
| Chaco Riparian Marsh and Shrubland | 39.1% | 55.4% | - | - | - | - | - | - |
| Chaco Xeromorphic Cliff and Other Rock Vegetation | 7.8% | 83.0% | - | - | - | - | - | - |
| Chaco-Espinal Brackish Marsh | 43.9% | 73.3% | - | - | - | - | - | - |
| Choco-Darien Floodplain Forest | - | - | 18.3% | 24.3% | 32.8% | 77.2% | <0.1% | - |
| Guajiran Flooded Forest | - | - | 56.5% | 61.1% | - | - | - | - |
| Guajiran Seasonal Dry Forest | - | - | 60.3% | 46.0% | - | - | - | - |
| Guajiran Xeromorphic Scrub and Woodland | - | - | 59.6% | 79.3% | - | - | - | - |
| Guayaquil Flooded and Swamp Forest | - | - | - | - | 49.4% | 56.2% | 36.9% | - |
| Llanos Flooded and Swamp Forest | - | - | 20.7% | 61.0% | - | - | - | - |
| Mesoamerican Coastal Plain Swamp Forest | - | - | 36.9% | - | 74.0% | - | 71.4% | - |
| Mesoamerican Floodplain Forest | - | - | 52.8% | 10.0% | - | - | - | - |
| Mesoamerican Freshwater Marsh, Wet Meadow and Shrubland | - | - | 41.8% | <0.1% | 54.2% | <0.1% | - | - |
| Mesoamerican-South American Pacific Coastal Salt Marsh | - | - | 25.0% | 100.0% | 59.3% | - | 38.2% | <0.1% |
| Neotropical Floating and Submerged Freshwater Marsh | 22.9% | 42.2% | 24.4% | - | 60.4% | - | 24.3% | 39.7% |
| Orinoquian Floodplain Peat Meadow and Marsh | - | - | 13.2% | 100.0% | - | - | - | - |
| Southern Andean Montane Salt Marsh | 87.5% | 87.5% | - | - | - | - | - | - |
| Southwestern Amazon Floodplain Forest | 7.7% | 7.6% | 81.1% | - | - | - | 2.7% | 5.2% |
| Tumbes Guayaquil Seasonal Dry Forest | - | - | 15.6% | <0.1% | 57.6% | 29.5% | 14.2% | 10.7% |
| Tumbesian Xeromorphic Scrub and Woodland | - | - | <0.1% | - | 53.9% | 20.6% | 29.9% | 10.0% |
| Western Ecuadorian Humid Forest | - | - | 37.8% | 9.4% | 59.0% | 41.2% | 58.6% | - |

Following broader range-wide patterns of loss, macrogroups with relatively northern distributions appear prominently in Table 2, with the Guajiran Xeromorphic Scrub and Woodland, the Guajiran Flooded Forest, the Mesoamerican Coastal Plain Swamp Forest, the Choco-Darien Floodplain Forest, and the Llanos Flooded and Swamp Forest, among the more characteristic types within Colombia and Ecuador with >50% or >70% loss within each country. Several types listed in Table 1 with a high percentage of loss within the Tropical Andes hotspot naturally occurred at lower elevations along the margins of the mapped hotspot boundary. Further south, macrogroups such as the Chaco Riparian Marsh and Shrubland and the Southern Andean Montane Salt Marsh were among those with the highest long-term loss (Table 2).

We then visualized these loss estimates per type within our study area by applying that loss estimate to the potential distribution map to depict distributions in terms of loss classes (>70% loss down to <10% loss) (Figure 4).

### 3.2. Macrogroup Representation in Protected Areas

Of the macrogroups in the study area, the greatest estimated proportion protected (i.e., 30–62%) was concentrated in either the northern or eastern extreme (e.g., the Western Atlantic and Caribbean Mangrove, the Central Guianan Montane Humid Forest, the Central Guianan Flooded Savanna).

Within the Tropical Andes countries, an average of only 15.4% of the pre-industrial extent and 17.8% of the recent extent of IVC macrogroups combined fell within protected areas (Figure 4). Higher proportions for ecosystem representation were mostly in types falling outside the Andes, as only 5.2% of the pre-industrial macrogroup extent within the Tropical Andes hotspot fell within protected areas (Supplementary Materials S2b). Only 22 of 126 macrogroups had more than 30% of recent extent protected, and only 10 had >50% of their recent extent in protected areas (Figure 4, Supplementary Materials S2b). However, in the Andes hotspot, only five of 95 had more than 30% of their extent in protected areas, and none had >50% protected (Figure 4, Table 2). For macrogroups occurring across the

four countries, 20 were protected in >30% of their pre-industrial extent, and only five were protected in >50% (Figure 5, left).

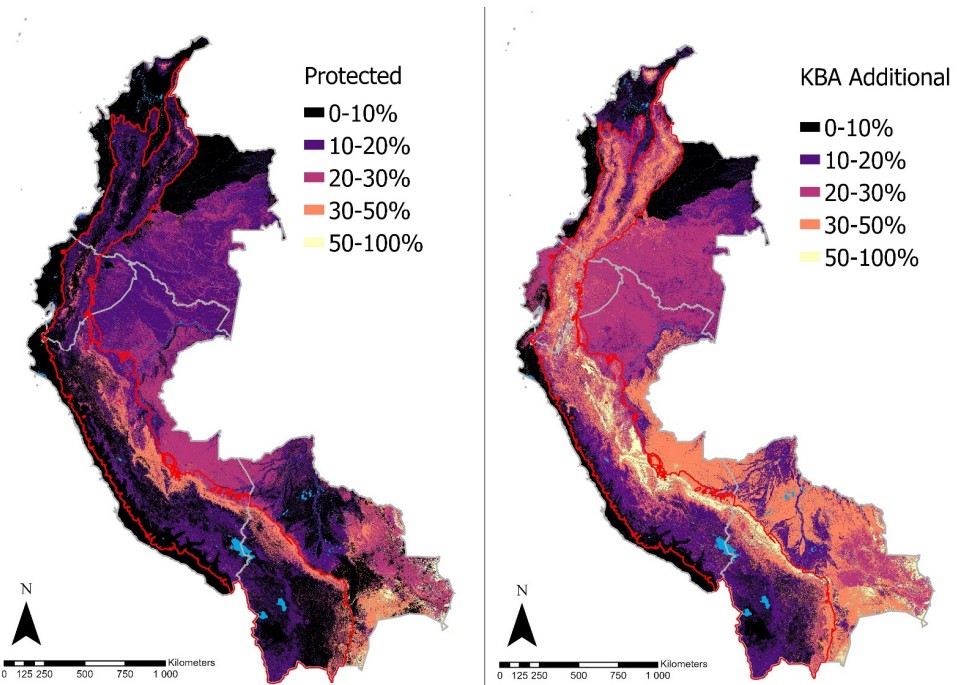

**Figure 5.** Proportional representation protected in IUCN I-VI lands by IVC macrogroup (**left**) and (**right**) proportion protected with the additional KBAs by IVC macrogroup (n = 126).

Country summaries are found in Supplementary Materials S2a. That table includes 104 macrogroups where protected percentages include those with <5% or 25–30% within a given country. This table highlights substantial gaps in protection as well as instances where low protection is evident in one country while moderate to high protection is evident for the same macrogroup in adjacent countries. Seventy-two of 104 macrogroups (69%) indicated only low levels of protection within any Tropical Andes country, allowing us to identify potential targets for increasing representation of ecosystem diversity. Examples of under-represented types occurring within the Tropical Andes hotspot include the Andean Puna Wet Meadow, the Bolivian-Tucuman Seasonal Dry Forest, the Central Andean Altiplano Salt Flats, the Central Andean (Yungas) Upper Montane Grassland and Shrubland, the Northern Andean Seasonal Dry Forest, and the Northern Andean Xeromorphic Scrub and Woodland (Supplementary Materials S2b).

Those macrogroups from the Tropical Andes hotspot that are represented well in some countries but poorly in others were the Bolivian-Tucuman Montane Grassland and Shrubland, the Central Andean (Yungas) Montane and Upper Montane Humid Forest, the Central Interandean Xeromorphic Scrub and Grassland, the High Northern Andean Super-Paramo, the Northern Andean Paramo, and the Northern Andean Xeromorphic Scrub and Woodland (Supplementary Materials S2b).

### 3.3. Additional Macrogroup Representation in Key Biodiversity Areas

Additional proportions of each IVC macrogroup (pre-industrial extent) that could be secured through identified Key Biodiversity Areas within the Tropical Andes countries can be visualized in Figure 5 (right). Supplementary Materials S2b summarizes the results for 112 of the 126 macrogroups occurring across one or more of the Tropical Andes countries, each with <30% range-wide representation within existing protected areas.

With the addition of KBAs, 16 macrogroups in Bolivia (e.g., the Beni Flooded Savanna, the High Andean Moist Puna Bunch Grassland), four in Colombia (e.g., the Catatumbo Magdalena Humid Forest), 21 in Ecuador (e.g., the Western Amazon Floodplain Forest,

the Central Andean (Yungas) Montane and Upper Montane Humid Forest), and 11 in Peru (e.g., the Eastern Subandean Ridge Montane Humid Forest, the Central Andean (Yungas) Lower Montane Humid Forest) would cross the >30% threshold of representation of under-represented types in either the country or the Tropical Andes hotspot, respectively.

### 3.4. Utilizing the IVC Classification Hierarchy for EBV Roll-Up

The IVC hierarchical classification structure offers several opportunities for linking these measures to a variety of common uses in environmental conservation. Essential Biodiversity Variables for ecosystem structure, including trends in ecosystem loss and/or proportion protected, can utilize maps depicting the IVC taxonomic hierarchy (Table 1) to "roll-up" measurements to continental or global scales. We illustrate this roll-up function for IVC macrogroup results in our study area by mapping the formation level of IVC (level 3) corresponding to a total number of 19 distinct formations in the region (Figure 6). IVC Formations are expressions of vegetation types with global physiognomy, i.e., responding to global patterns of climate, such as those found along latitude and elevation gradients. Given that they do not express floristic composition at continental scales, they are most readily linked to more global-scaled land cover maps, so trends analysis in the Tropical Andes using these vegetation maps may be readily linked to global trend assessments. These more generalized results depict overall patterns expressed by the 126 IVC macrogroups, although they necessarily mask many more localized patterns (e.g., compared with Figures 4 and 5).

Table 3 summarizes these results by country, indicating, for example, where long-term loss is highest among Mangrove, Salt Marsh, Tropical Dry Forest and Woodland, and Warm Desert and Semi-Desert Scrub and Grassland within some or all of these four Tropical Andes countries. While the protected percentage is well-illustrated here by IVC Formation within each country, the additional proportion protected by securing KBAs appears to be somewhat less sensitive using this classification scale.

**Table 3.** Long-term loss in extent, percentage protected, and percentage secured in KBAs for IVC Formations summarized by country and within the Tropical Andes hotspot portion of each country (green colors = least concern; red colors = most concern).

| IVC Formations | Boliva Loss % | Boliva Protected % | Boliva Protected with KBA % | Peru Loss % | Peru Protected % | Peru Protected with KBA % | Ecuador Loss % | Ecuador Protected % | Ecuador Protected with KBA% | Colombia Loss % | Colombia Protected % | Colombia Protected with KBA % |
|---|---|---|---|---|---|---|---|---|---|---|---|---|
| Cool Semi-Desert Scrub and Grassland | 8% | 4% | 6% | 10% | 12% | 13% | | | | | | |
| Mangrove | | | | 49% | 17% | 17% | 47% | 24% | 24% | 23% | 13% | 13% |
| Mediterranean Scrub and Grassland | 2% | 5% | 5% | 0% | 0% | 0% | | | | | | |
| Salt Marsh | 9% | 2% | 3% | 19% | 9% | 10% | 59% | 7% | 7% | 37% | 5% | 5% |
| Tropical Bog and Fen | 21% | 19% | 28% | 29% | 19% | 24% | 49% | 28% | 42% | 39% | 33% | 47% |
| Tropical Cliff, Scree, and Other Rock Vegetation | 25% | 14% | 14% | 14% | 2% | 20% | 23% | 13% | 39% | 41% | 8% | 19% |
| Tropical Dry Forest and Woodland | 22% | 27% | 28% | 15% | 11% | 25% | 50% | 5% | 14% | 49% | 5% | 8% |
| Tropical Flooded and Swamp Forest | 12% | 19% | 19% | 4% | 12% | 14% | 23% | 33% | 35% | 15% | 16% | 17% |
| Tropical Freshwater Aquatic Vegetation | 23% | 15% | 15% | 24% | 1% | 3% | 60% | 3% | 3% | 24% | 14% | 14% |
| Tropical Freshwater Marsh, Wet Meadow, and Shrubland | 12% | 11% | 12% | 9% | 10% | 12% | 20% | 27% | 32% | 19% | 9% | 9% |
| Tropical High Montane Scrub and Grassland | 7% | 12% | 18% | 12% | 11% | 14% | 18% | 51% | 60% | 12% | 45% | 54% |
| Tropical Lowland Grassland, Savanna, and Shrubland | 22% | 50% | 50% | 7% | 22% | 32% | 9% | 23% | 23% | 17% | 16% | 16% |
| Tropical Lowland Humid Forest | 5% | 31% | 31% | 3% | 20% | 25% | 26% | 16% | 19% | 13% | 20% | 21% |
| Tropical Montane Grassland and Shrubland | 18% | 14% | 25% | 20% | 5% | 13% | 36% | 30% | 47% | 31% | 28% | 42% |
| Tropical Montane Humid Forest | 13% | 29% | 37% | 10% | 17% | 35% | 22% | 18% | 47% | 29% | 14% | 24% |
| Tropical Thorn Woodland | 19% | 27% | 30% | 21% | 5% | 13% | 40% | 8% | 21% | 48% | 6% | 13% |
| Warm Desert and Semi-Desert Scrub and Grassland | 34% | 21% | 22% | 10% | 3% | 5% | 39% | 2% | 39% | 46% | 5% | 15% |

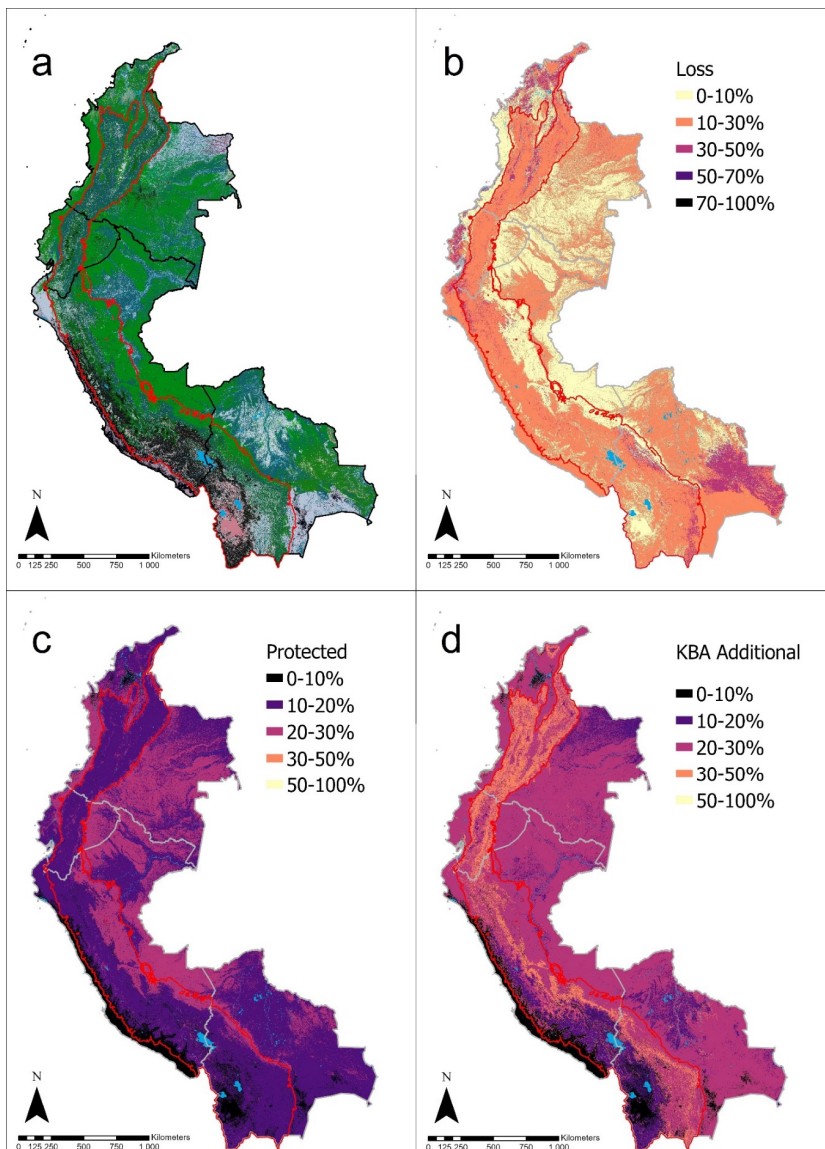

**Figure 6.** (**a**) 19 IVC Formations. (**b**) Long—term proportional loss in extent, (**c**) proportional representation, and (**d**) additional protection with KBAs.

### 3.5. Utilizing the IVC for EBV Step-Down for Local Uses

As partly illustrated above, IVC macrogroups can be quite helpful for regional and national assessment and decision support. However, many resource managers at more local scales will prefer to work with mapped distributions of ecosystems described at finer thematic resolutions. One example of this from our study area is the consolidated map products using the NatureServe terrestrial ecological systems classification [30,31]. These units approximate the scale and concept of IVC group at level 6 of that hierarchy (Table 1). As compared with 126 IVC macrogroups in this study area, there was a total of 330 level 6 equivalent types in the countries of the Andes with 209 of those within the Tropical Andes hotspot (Figure 7). Figure 8 depicts side-by-side examples of the mapped concepts used in this study, with the thematically coarsest Formation (IVC Level 3) on the left, the macrogroup (IVC Level 5) in the center, and the Group equivalent (IVC Level 6) on the right. Methods for mapping differ between Level 3/Level 5 examples and the Level 6 example, but this figure provides an indication of these varying levels of thematic detail as they occur across this highly varied regional landscape.

Natural resource planners and managers at more local scales can utilize these maps to characterize ecological gradients, map habitats for at-risk species, document ecological

conditions or integrity, and take conservation actions involving vegetation treatments within and among protected areas [32–34]. Therefore, the hierarchical structure of the IVC, when applied to map production across the Tropical Andes, offers ample opportunities to link vegetation observations with remotely sensed data [25,26,31] to map vegetation and land cover at multiple spatial and thematic resolutions. These maps in turn enable reporting of Essential Biodiversity Variables for users working from local to regional to global scales.

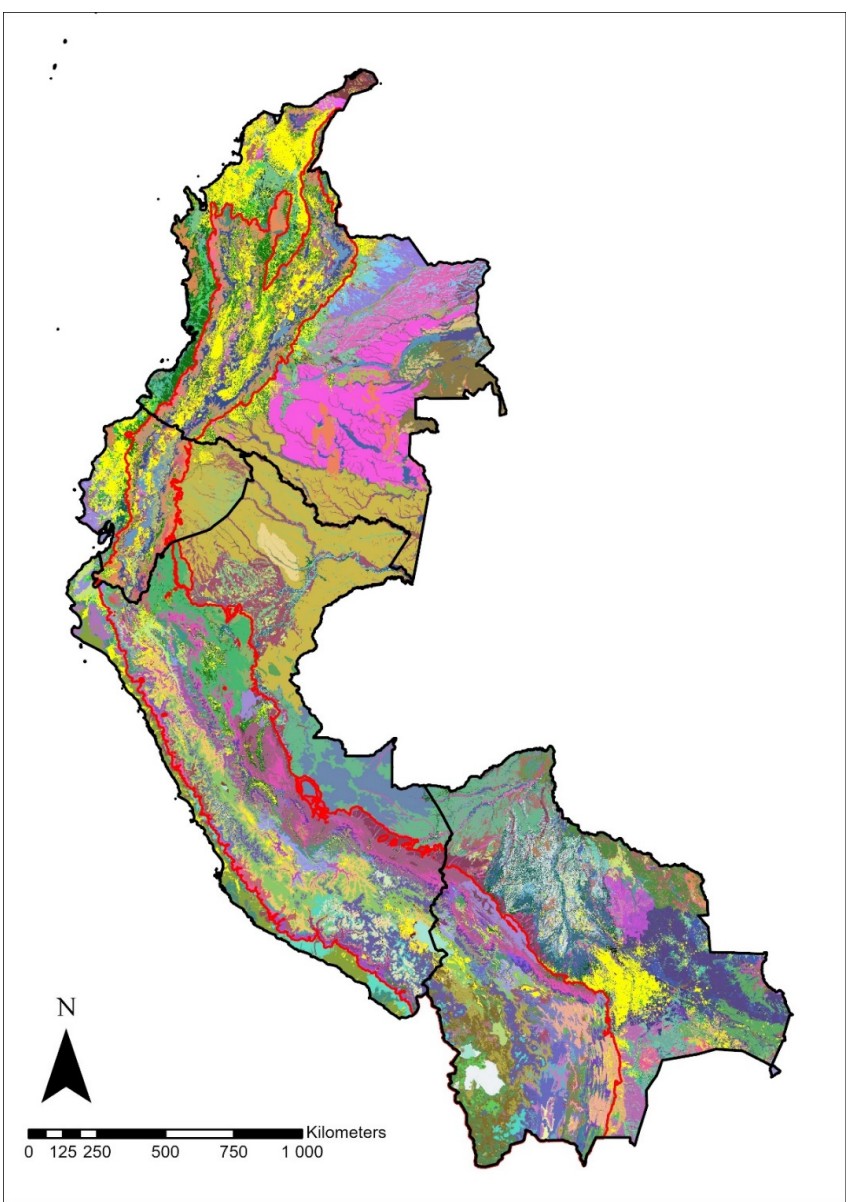

**Figure 7.** Distribution of natural ecosystem types and human-dominated land uses (bright yellow) approximating L6 of the IVC classification hierarchy.

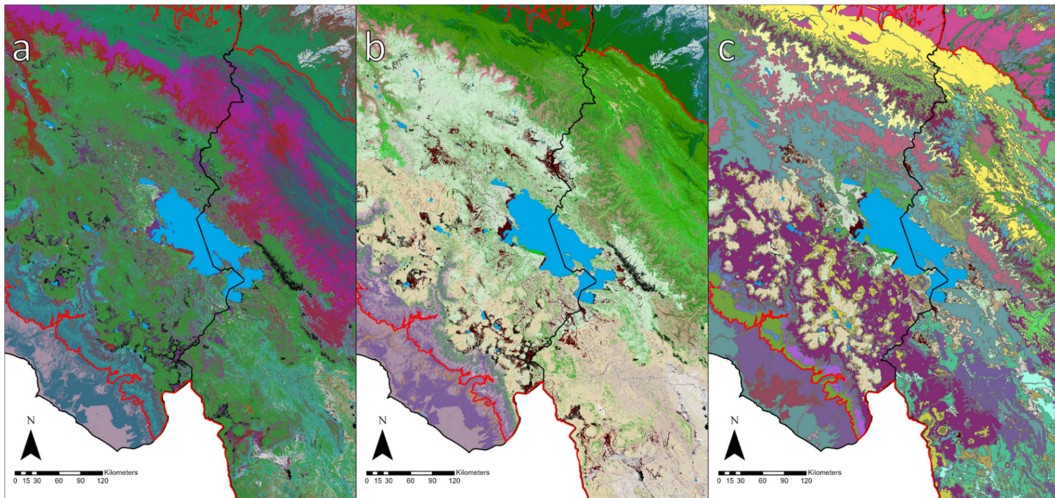

**Figure 8.** Cross-section of ecosystem types within southern Peru/Bolivia displayed at three levels of the IVC classification hierarchy. (**a**) Formation, (**b**) Macrogroup, and (**c**) Group.

## 4. Discussion

Documenting trends in ecosystem extent and ecosystem diversity that occur under conservation-oriented land management are both essential to monitoring policy to biodiversity conservation goals. Map products provided in this study across thematic and spatial resolutions provide a powerful tool for prioritizing conservation in landscapes where urgent conservation action is needed, whether for land or ecosystem protection. We also can reflect on the process of identifying these indicators, and the additional opportunities presented by our approach.

### 4.1. Applying the IVC Classification Hierarchy from Local to Global Scales

The IVC hierarchical classification structure facilitates linking measures of ecosystem diversity across scales of conservation action. We illustrate where regional assessment at the scale of the Tropical Andes can provide insight for regional conservation investments and be readily scaled up for reporting at continental or global scales. At the same time, these same measures may be linked to ecosystem concepts defined and mapped for focused attention by land-use planners and managers working at more local scales. These mapped concepts can be readily linked to national land cover products currently in use locally, as in Colombia [35].

### 4.2. Lesson Learned from the EBVs Workshop

In addition to the technical design aspect of estimating the ecosystem extent at two time steps (pre-industrial and current) and deriving a set of EBV-based indicators, this study highlights the co-design process of bringing together policy and decision-makers with users of biodiversity information to better understand their needs. The process was based on one of the most important steps of the methodology applied in the Tropical Andes for the design and implementation of a Biodiversity Observation Network for assessing user needs [24]. Although it is vital to engage with relevant stakeholders in all sectors to successfully respond to current biodiversity challenges, the scientific and technical communities producing biodiversity information rarely engage with or fully recognize the needs of users of biodiversity information. Additionally, the information produced by researchers, which is critical for decision making, remains inaccessible to those outside of academia due to the use of highly specialized and incomprehensible language of scientific results. Directly engaging with and having highly technical discussions with relevant individuals who do not have scientific training but are trained in the area of public policy should be replicated. This model can further make biodiversity information useful to those that need it to help achieve a more sustainable world. Employing feasible and repeatable

analyses such as this can not only strengthen the visibility and critical nature of biodiversity data for decision-making but also motivate targeted efforts to strengthen biodiversity monitoring in the region.

## 5. Conclusions

Mapped distributions of the International Vegetation Classification, both "pre-industrial" and "current", provided important opportunities for assessing trends in ecosystem extent and condition. We have demonstrated novel approaches to using these data as Essential Biodiversity Variables that respond directly to the needs expressed by regional stakeholders.

They offered important contributions to prioritizing lands for enhanced conservation attention under global and national representation targets under the Convention on Biological Diversity [21], to documenting the relative conservation status of ecosystems, such as with the IUCN Red List of Ecosystems [36,37], and for identifying Key Biodiversity Areas or other area-based conservation measures (OECM). Of course, we acknowledge that these data need to be augmented with local information—much of which could only be gathered in the field—to implement conservation actions on the ground. We also want to acknowledge that lands prioritized for enhanced conservation will often include areas where people live and work, so clear mechanisms are needed to support compatible and truly sustainable land management.

**Supplementary Materials:** The following supporting information can be downloaded at: https://www.mdpi.com/article/10.3390/rs14122847/s1, S1_IVC Macrogroups_Rangewide.xlsx includes links to type descriptions and range-wide information on long-term loss and protection. S2_IVC Macrogroups_AndesCountries.xlsx includes statistics on each type as derived for this analysis.

**Author Contributions:** Conceptualization, P.J.C., M.F. and J.V.; formal analysis, J.V.; writing—original draft preparation, P.J.C., J.V. and M.F.; methodology, P.J.C., M.F., J.V., H.M.P., C.A.-M., F.C., F.J.B.G., X.C., L.C., F.D., E.Y.D.R., M.G., C.J., I.L.C., R.L., D.L.-A., A.M., G.M.J., L.N., S.N., F.P.-A., G.R.O., M.F.T. and C.Z.-T.; writing—review and editing, P.J.C., M.F, J.V., H.M.P., C.A.-M., F.C., F.J.B.G., X.C., L.C., F.D., E.Y.D.R., M.G., C.J., I.L.C., R.L., D.L.-A., A.M., G.M.J., L.N., S.N., F.P.-A., G.R.O., M.F.T. and C.Z.-T.; visualization, P.J.C., J.V. and M.F.; project administration, P.J.C. All authors have read and agreed to the published version of the manuscript.

**Funding:** This research received funding from the ERANet Joint Call 2016–2017 (DLR Förderkennzeichen 01DN19032 Tropical Andes Observatory—TAO).

**Data Availability Statement:** Data is accessible from the GEOBON EBV portal at the macrogroup (https://doi.org/10.25829/xe5m37) and formation (https://doi.org/10.25829/c8rz53) level of the International Vegetation Classification (IVC).

**Acknowledgments:** We thank all 2019–2022 workshop participants from several local institutions in the Tropical Andes, including Conservacion Amazonica-Peru (ACCA), Asociacion Boliviana para la Investigacion de Ecosistemas Andino Amazonicos (ACEAA), Fundacion Ecociencia, Ecuador, Instituto Nacional de Biodiversidad de Ecuador (INABIO), and several international institutions including NatureServe, Universidad de Cordoba in Spain, the Global Biodiversity Information Facility (GBIF), the German Centre for Integrative Biodiversity Research (iDiv), and the Group on Earth Observations—Biodiversity Observation Network (GEO BON) who worked together to document needs of biodiversity data users in the Tropical Andes.

**Conflicts of Interest:** The authors declare no conflict of interests.

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
