# Peer review of "Conserving Ecosystem Diversity in the Tropical Andes"

_remotesensing, doi:10.3390/rs14122847_

Round 1

Reviewer 1 Report

The establishment of protected areas is a key strategy for biodiversity conservation. Over the years a variety of criteria, approaches, and data sources, have been used to set aside areas for conservation. This together with local, regional, and global needs reflects the complexity involved in identifying important areas, and ultimately understanding threats and conservation needs. Focusing on “ecologically representative” areas in the highly diverse Tropical Andes, this paper examines the degree of representation of ecological units (macrogroups from the IVC classification system) and Key Biodiversity Areas (KBC) in the protected areas of this region.  Using pre-industrial and recent macrogroup maps, the authors establish the extent of loss among the macrogroups, information that can be used to set conservation targets.

General comments:

  • Regional analyses such as the one described in this paper entail are challenging for numerous regions. The general approach described in this paper is interesting in that it starts from maps that used “occurrences” of macrogroups to model their potential and recent distribution.
  • There is no perfect vegetation classification system, and it would be worth discussing how other efforts have been used in similar work. For example, In Colombia a modified version of Corina is used for conservation planning purposes (https://www.researchgate.net/publication/303960063_LEYENDA_NACIONAL_DE_COBERTURAS_DE_LA_TIERRA_METODOLOGIA_CORINE_LAND_COVER_ADAPTADA_PARA_COLOMBIA_ESCALA_1100000/link/5760209008ae227f4a3eee08/download).
  • The paper can benefit from a better organization, and clarification in certain areas.
  • The Discussion seems to include information more appropriate for the Results. For example, starting in Line 370 the authors present similar analyses but at a higher level of the classification hierarchy.

Specific comments:

  • Line 119. The classification system is of ecosystems? Vegetation types? Macrogroups? Use same terminology through paper. For example, Line 166 refers to Vegetation types, line 165 to “Target Ecosystems”, and Line 216 to “Macrogroups.” A single sentence reading “From now on we refer to xxxx as macrogroups” could help, and then from there keep using the chosen word.
  • Line 166. Is the International Vegetation Classification (IVC) used worldwide or only in certain regions? I would be useful to know the extent to which it has been adopted.
  • Line 169. Table 1 should add a column listing type of agency that has adopted the various levels or whose work is based on those levels. For example, in Line 169 it reads that “follows that described as a federal standard for vegetation description in the United States.”
  • Line 173. Perhaps it would be useful to have the names of the various categories that each level can take.
  • Line 173. It seems that “Class” and “Subclass” levels could be renamed into Global Life form Physiognomy and Sub-global Life form Physiognomy, the latter denoting something narrower. Idem for the next level.
  • Line 177. Since the vegetation macrogroup level is central to this work it is important to better characterize it. It seems that at this level life form, latitude, elevation, humidity, and regions are used to define the macrocroups. The region level is not clear.
  • Line 182. It would be worth describing the general mapping approach used to generate the “potential” and “recent” maps. Perhaps a table showing the needed data for each? For example, lines 188-190 should be part of Line 183.
  • Line 187. Mapping resolution 270 m. Why is this? What data source has this resolution? This appears in line 205. It is better to describe what is behind the maps in this section
  • Line 192. Given that these are modelled vegetation types, has there been any type of verification?
  • Supplementary Appendix 1. I would organize it a little bit different. I would put columns I, H, G, F after column A.
  • Figure 2. I could be nice to have a C panel showing the difference between B and A.
  • Line 203. Part of the material that appears here should be part of the previous section.
  • Line 216. It seems that this section is critical to the paper. Therefore, I would give more strength to the first section with something like “To calculate/estimate the proportion of each macrogroup in existing protected areas …”
  • Line 216. The source for protected areas has not been indicated. How does this source compare to country level listings of protected areas?
  • Line 227. This is the first time that KBAs are mentioned in this paper. This should be included throughout the paper.
  • Line 243. Perhaps start this section with the general findings, and then more specific ones?
  • Line 238. This section can be better organized – start from general trends and then go into specifics.
  • Line 253. The word hotspot appears for the first time. Does this refer to the KBA?
  • Figure 4 may become panel C in Figure 2.
  • Line 277. This is a long sentence that is hard to understand.
  • Line 282. This is a general sentence that may be used in Methods or to open this section to warn the readers of the problems with protection in protected areas.
  • Line 291. Andes hotspot. What does this mean? Is the entire Andes or one of the KBA regions?
  • Line 315. Is there an analysis of KBA regions within the protected areas?
  • Figure 5 is confusing because it is a continuous map, and the KBA’s are discontinuous (Figure 3?)

Author Response

General comments:

  • Regional analyses such as the one described in this paper entail are challenging for numerous regions. The general approach described in this paper is interesting in that it starts from maps that used “occurrences” of macrogroups to model their potential and recent distribution.

Response – yes, one can see details of mapping methods in citations 25 and 26.

  • There is no perfect vegetation classification system, and it would be worth discussing how other efforts have been used in similar work. For example, In Colombia a modified version of Corina is used for conservation planning purposes (https://www.researchgate.net/publication/303960063_LEYENDA_NACIONAL_DE_COBERTURAS_DE_LA_TIERRA_METODOLOGIA_CORINE_LAND_COVER_ADAPTADA_PARA_COLOMBIA_ESCALA_1100000/link/5760209008ae227f4a3eee08/download).

Response – we agree that there is no perfect classification, as they all aim to address various purposes and reflect expert knowledge and methods of the time. We are familiar with the Corine classification as it was consulted along with many others in the development of the classification used in this study. We cited IDEAM 2010 in the Conclusions section. For this paper, we more simply point out the value of a hierarchically structured classification that has been mapped across the continent, and how that could contribute toward measurable variables for ecosystem conservation.

  • The paper can benefit from a better organization, and clarification in certain areas.

Response – we appreciate suggestions below along these lines, and have implemented adjustment accordingly.

  • The Discussion seems to include information more appropriate for the Results. For example, starting in Line 370 the authors present similar analyses but at a higher level of the classification hierarchy.

Response – We agree. We moved this content to the results section, and introduced the multi-level analysis in the methods section.

Specific comments:

  • Line 119. The classification system is of ecosystems? Vegetation types? Macrogroups? Use same terminology through paper. For example, Line 166 refers to Vegetation types, line 165 to “Target Ecosystems”, and Line 216 to “Macrogroups.” A single sentence reading “From now on we refer to xxxx as macrogroups” could help, and then from there keep using the chosen word.

Response – Throughout the manuscript we have used the term “ecosystem” generically with a specific hierarchically structured vegetation classification (the IVC) to define the units of mapping and analysis. We refer to macrogroups for most of the paper; as those are the central level of classification used in the analysis However, we also needed to leave room for reference to other classification levels in the Discussion section. We have rechecked the text and made adjustments for clarity.

  • Line 166. Is the International Vegetation Classification (IVC) used worldwide or only in certain regions? I would be useful to know the extent to which it has been adopted.

Response – see associated citations (25-27). The upper levels of the IVC hierarchy are global. The  IVC itself has been used most extensively across the Americas, but there is expanding interest in integration with locally used systems in Europe and Australia. It has also been used cross the African continent (Sayre et al. Sayre, R., et al.2013. A New Map of Standardized Terrestrial Ecosystems of Africa. Washington, DC: Association of American Geographers. )

  • . Line 169. Table 1 should add a column listing type of agency that has adopted the various levels or whose work is based on those levels. For example, in Line 169 it reads that “follows that described as a federal standard for vegetation description in the United States.”

Response – While we appreciate the comment, issues of who use the classification for which purposes is covered in part in citation 27 (at least as it stood in 2014) but would be a distraction for the intent of this manuscript. Our methods demonstrated here could be replicated with other hierarchically structured and mapped classifications around the world.

  • Line 173. Perhaps it would be useful to have the names of the various categories that each level can take.

Response – Names of examples were included in the table. That has been made clearer with inserted text.

  • Line 173. It seems that “Class” and “Subclass” levels could be renamed into Global Life form Physiognomy and Sub-global Life form Physiognomy, the latter denoting something narrower. Idem for the next level.

Response – We understand the comment. These are simply defining characteristics defined in citation (27) for IVC Class and Subclass levels; the Class based on universal life form physiognomy (e.g., “Forest & Woodland”) and the Subclass being one variant found around the globe “Tropical Forest and Woodland”)

  • Line 177. Since the vegetation macrogroup level is central to this work it is important to better characterize it. It seems that at this level life form, latitude, elevation, humidity, and regions are used to define the macrocroups. The region level is not clear.

Response – At line 180 we inserted another reference to citation 27 for much in-depth explanation of the IVC, including macrogroup concepts.  One can also review Appendix 1 for detailed listing and links to descriptions of all macrogroups involved in the study.

  • Line 182. It would be worth describing the general mapping approach used to generate the “potential” and “recent” maps. Perhaps a table showing the needed data for each? For example, lines 188-190 should be part of Line 183.

Response – Please see citation 25 for in depth explanation of mapping methods. Since this manuscript aims primarily to illustrate the use of these maps with other data and purposes, we felt it best to simply cite those sources for readers interested in those methods and data.

  • Line 187. Mapping resolution 270 m. Why is this? What data source has this resolution? This appears in line 205. It is better to describe what is behind the maps in this section

Response - Please see citation 25 for in depth explanation of mapping methods. Since this manuscript aims primarily to illustrate the use of these maps with other data and purposes, we felt it best to simply cite those sources for readers interested in those methods and data.

  • Line 192. Given that these are modelled vegetation types, has there been any type of verification?

Response – Yes there was. Please see citation 25 for in depth explanation of mapping methods. We added text referencing validation statistics that can be found in citation 25. Since this manuscript aims primarily to illustrate the use of these maps with other data and purposes, we felt it best to simply cite those sources for readers interested in those methods and data.

  • Supplementary Appendix 1. I would organize it a little bit different. I would put columns I, H, G, F after column A.

Response – We appreciate the comment. However, we chose to structure the table from left to right, with columns A-D applicable to macrogroups, then extending to higher classification levels further to the right.

  • Figure 2. I could be nice to have a C panel showing the difference between B and A.

Response - Figure 2 was just intended to communicate the pre-industrial distributions, while Figure 4 intended as a form of visualizing long-term proportional loss, so we felt it best to keep them separate.

  • Line 203. Part of the material that appears here should be part of the previous section.

Response – We agree. Reference to mapping current land use was moved up to line 188-191.

  • Line 216. It seems that this section is critical to the paper. Therefore, I would give more strength to the first section with something like “To calculate/estimate the proportion of each macrogroup in existing protected areas …”

Response – We implemented this suggestion.

  • Line 216. The source for protected areas has not been indicated. How does this source compare to country level listings of protected areas?

Response – add date for accessing the WDPA data in text.  We also added text indicating that “We did not compare this global data set to country-specific protected area data sets for this analysis.”

  • Line 227. This is the first time that KBAs are mentioned in this paper. This should be included throughout the paper.

Response – KBAs are mentioned first in the abstract, and then in methods. We added further text in the introduction to clarify what KBAs are.

  • Line 243. Perhaps start this section with the general findings, and then more specific ones?

Response – Text was modified to simplify and clarify these results.

  • Line 238. This section can be better organized – start from general trends and then go into specifics.

Response – We agree. We adjusted text accordingly.

  • Line 253. The word hotspot appears for the first time. Does this refer to the KBA?

Response – it appears first at line 134 and adds citation #1.

  • Figure 4 may become panel C in Figure 2.

Response – Figure 2 was just intended to communicate the pre-industrial distributions, while Figure 4 intended as a form of visualizing long-term proportional loss, so we felt it best to keep them separate.

  • Line 277. This is a long sentence that is hard to understand.

Response – this sentence was revised for clarity: “Of the macrogroups in the study area, the greatest estimated proportion protected (i.e., 30-62%) is concentrated in either the northern or eastern extreme (e.g., Western Atlantic & Caribbean Mangrove, Central Guianan Montane Humid Forest, Central Guianan Flooded Savanna).”

  • Line 282. This is a general sentence that may be used in Methods or to open this section to warn the readers of the problems with protection in protected areas.

Response – We agree. We moved that sentence up to around Line 222 in Methods.

  • Line 291. Andes hotspot. What does this mean? Is the entire Andes or one of the KBA regions?

Response – a link is provided in Figure 1 caption for background on the area and its delineation as a global biodiversity “hotspot.” Critical Ecosystem Partnership Fund

  • Line 315. Is there an analysis of KBA regions within the protected areas?

Response – We presume the reviewer means the ecosystem representation within the KBA vs. the existing protected area. These data were generated, and are found in Appendix 2, but we chose to highlight the additional area that would be conserved within the KBAs, accounting for portions where the proposed KBAs overlap existing protected areas.

  • Figure 5 is confusing because it is a continuous map, and the KBA’s are discontinuous (Figure 3?)

Response – the map depicts the effect of conserving KBAs in terms of additional ecosystem representation (as expressed in the continuous color ramp).

Reviewer 2 Report

This is a useful analysis of a very large, remote and diverse region.  As such, it represents a new look at threat classification using existing datasets.  I think the work should be published and the graphics are of good quality.  But, while the authors did point out a few issues with the methods, I think they should make  a bigger point about limitations of this approach for many conservation questions.  That is, although the work is very useful for broad analysis, myriad aspects of biodiversity cannot be captured by remote methods (e.g. species diversity of any taxa, presence/abundance of apex predators, some forms of forest degradation, etc.), which will put some limits on the application of such approaches for planners managers in the field. For many conservation questions, there is simply no escape from field work.

In short, I commend them for their great efforts and the fine quality of the writing and the tables/figures, and I agree the work is important, but I think they need to add a few more obvious caveats about the utility and limitations of this approach.

Author Response

This is a useful analysis of a very large, remote and diverse region.  As such, it represents a new look at threat classification using existing datasets.  I think the work should be published and the graphics are of good quality.  But, while the authors did point out a few issues with the methods, I think they should make  a bigger point about limitations of this approach for many conservation questions.  That is, although the work is very useful for broad analysis, myriad aspects of biodiversity cannot be captured by remote methods (e.g. species diversity of any taxa, presence/abundance of apex predators, some forms of forest degradation, etc.), which will put some limits on the application of such approaches for planners managers in the field. For many conservation questions, there is simply no escape from field work.

In short, I commend them for their great efforts and the fine quality of the writing and the tables/figures, and I agree the work is important, but I think they need to add a few more obvious caveats about the utility and limitations of this approach.

Response – We appreciate the comments and certainly agree that this is analysis should not presume to account for all information – much of which could only be gathered in the field – to implement effective conservation. We stated our objectives for this type of analysis in the Introduction, and then highlighted the multi-scale implications of these data in the Discussion, including use of finer-scale thematic and spatial ecosystem classifications that have proven their utility for on-the-ground conservation. We mainly intend to illustrate one pathway to link data and analyses for conservation decisions across multiple scales. We inserted language to this effect in the Conclusions. “Of course, we acknowledge that these data need to be augmented local information - much of which could only be gathered in the field – to implement conservation actions on the ground.”

Reviewer 3 Report

The analysis is based on previous remote sensing mapping of natural vegetation types (macrogroups), and losses of, and amount of protected areas in those types, in the Andes Biodiversity Hotspots.  Macrogroups in the vegetation classification system (274 types, 126 within the bounds of this study area) were analyzed, which seems to be an appropriate level within the vegetation classification system for this analysis, since the division level would not provide enough specific detail and the group level would have too many types to analyze (although this analysis is later placed into context of the higher and lower levels of classification).

More detailed maps and data (than is presented in the paper) for the macrogroups are provided at websites and in supplementary files. I can see that it is not possible to present more than a summary in the main paper, and to mention only those macrogroups with the largest level of loss since preindustrial times (e.g. table 2). It seems like a good idea to compare/analyze protected areas as well as key biodiversity areas (parts of which might not be in protected areas).

I agree with the first paragraph of the Discussion that this analysis serves as a good source of information for conservation planning in the Andes Hotspot, and about the lessons learned and comments about having a way to engaging local people in conservation plans.

The paper does a good job of showing the importance of the analysis in the context of higher and lower levels of the vegetation classification system—analysis at the formation level (2 steps above the macrogroup level) at a broader spatial scale (Figure 6 and table 3), as well as at the finer scales (i.e. the group level, Fig. 7).  Finally, there is an example of a comparison of the classification at three levels using a smaller spatial extent (formation, macrogroup and group, Fig. 8).

Here are some minor revisions to consider:

Abstract. Lines 43, 44 repetitive, mentions 'four countries' twice.

Line 46, define CBD—abstract needs to stand alone.

Line 126, end of introduction needs a more specific statement of objectives and/or questions to be answered. For example, to classify the vegetation types, loss of each vegetation type since pre-industrial times and amount of each included in protected areas and key biodiversity areas, and show how the classification is useful for conservation planning at different geographic extents and levels within the vegetation classification system (no doubt the authors can state this more accurately than I can).  

Author Response

The analysis is based on previous remote sensing mapping of natural vegetation types (macrogroups), and losses of, and amount of protected areas in those types, in the Andes Biodiversity Hotspots.  Macrogroups in the vegetation classification system (274 types, 126 within the bounds of this study area) were analyzed, which seems to be an appropriate level within the vegetation classification system for this analysis, since the division level would not provide enough specific detail and the group level would have too many types to analyze (although this analysis is later placed into context of the higher and lower levels of classification).

More detailed maps and data (than is presented in the paper) for the macrogroups are provided at websites and in supplementary files. I can see that it is not possible to present more than a summary in the main paper, and to mention only those macrogroups with the largest level of loss since preindustrial times (e.g. table 2). It seems like a good idea to compare/analyze protected areas as well as key biodiversity areas (parts of which might not be in protected areas).

I agree with the first paragraph of the Discussion that this analysis serves as a good source of information for conservation planning in the Andes Hotspot, and about the lessons learned and comments about having a way to engaging local people in conservation plans.

The paper does a good job of showing the importance of the analysis in the context of higher and lower levels of the vegetation classification system—analysis at the formation level (2 steps above the macrogroup level) at a broader spatial scale (Figure 6 and table 3), as well as at the finer scales (i.e. the group level, Fig. 7).  Finally, there is an example of a comparison of the classification at three levels using a smaller spatial extent (formation, macrogroup and group, Fig. 8).

Response – Thank you for your review and comments. Our intent was to illustrate data, methods, and visualizations at this regional scale to document some key aspects of ecosystem conservation, but also show how these data could be either scaled up for broader scales (like continental or global) or stepped down to local conservation applications. We moved this latter content to the results section, and introduced the multi-level analysis in the methods section.

Here are some minor revisions to consider:

Abstract. Lines 43, 44 repetitive, mentions 'four countries' twice.

Response - corrected

Line 46, define CBD—abstract needs to stand alone.

Response - corrected

Line 126, end of introduction needs a more specific statement of objectives and/or questions to be answered. For example, to classify the vegetation types, loss of each vegetation type since pre-industrial times and amount of each included in protected areas and key biodiversity areas, and show how the classification is useful for conservation planning at different geographic extents and levels within the vegetation classification system (no doubt the authors can state this more accurately than I can).  

Response – We inserted the current objective statement (lines 116-119) as follows: “Our objective is to use remote sensing data and analytical tools that facilitate the assessment of trends in ecosystem extent and proportional area protected building on the results of recent mapping efforts with hierarchical vegetation classification [25, 26].”